# Molecular Detection and Identification of *Chlamydiaceae* in the Eyes of Wild and Domestic Ruminant Hosts from Northern Spain

**DOI:** 10.3390/pathogens10030383

**Published:** 2021-03-23

**Authors:** Andrea Dias-Alves, Oscar Cabezón, Nicole Borel, Jorge Ramón López-Olvera, Gregorio Mentaberre, Santiago Lavín, Xavier Fernández Aguilar

**Affiliations:** 1Wildlife Conservation Medicine Research Group (WildCoM), Departament de Medicina i Cirurgia Animals, Universitat Autònoma de Barcelona (UAB), Bellaterra, 08193 Barcelona, Spain; andrea.da114@gmail.com (A.D.-A.); Oscar.Cabezon@uab.cat (O.C.); 2Institute of Veterinary Pathology, Department of Pathobiology, Vetsuisse Faculty, University of Zurich, Winterthurerstrasse 268, CH-8057 Zurich, Switzerland; nicole.borel@uzh.ch; 3Wildlife Ecology & Health Research Group (WE&H) and Servei d’Ecopatologia de Fauna Salvatge (SEFaS), Departament de Medicina i Cirurgia Animals, Facultat de Veterinària, Universitat Autònoma de Barcelona (UAB), Bellaterra, 08193 Barcelona, Spain; gregorio.mentaberre@udl.cat (G.M.); santiago.lavin@uab.cat (S.L.); 4Serra Húnter fellow, Wildlife Ecology & Health Group Research Group (WE&H), Departament de Ciència Animal, Escola Tècnica Superior d’Enginyeria Agraria (ETSEA), Universitat de Lleida (UdL), 25198 Lleida, Spain; 5Department of Ecosystem & Public Health, University of Calgary, Calgary, AB T2N 1N4, Canada; xfdezaguilar@gmail.com

**Keywords:** *Chlamydia pecorum*, domestic sheep, infectious keratoconjunctivitis, ocular disease, Pyrenean chamois, *Rupicapra pyrenaica*, wildlife–livestock interface

## Abstract

Infections by Chlamydiae are associated with ocular disease in humans and animals. In this study, the presence and diversity of *Chlamydia* spp. was assessed in diseased and healthy eyes of domestic sheep and wild ruminants that share mountain habitats in northern Spain. The presence of *Chlamydia* spp. was tested by real-time PCR in 1786 conjunctival swabs collected from both eyes of 893 animals from mountain habitats in northern Spain, and chlamydial species were identified in the positive samples by ArrayTube microarray methods. Chlamydial DNA was detected in 0.6% (CI95% 0.2–1.3) of the Pyrenean chamois (*Rupicapra pyrenaica*) and 1.4% (CI95% <0.01–8.1) of the sheep (*Ovis aries*) sampled, with *Chlamydia pecorum* the only chlamydial species identified. No association of *C. pecorum* with ocular disease or co-infection with *Mycoplasma conjunctivae* was found. Further studies on the pathogenesis of infectious keratoconjunctivitis are needed to better understand the ecology of *C. pecorum* and its possible role as a ruminant pathogen at the wildlife–livestock interface.

## 1. Introduction

Chlamydiae are obligate intracellular gram-negative bacteria affecting both humans and animals. According to the current taxonomy, the *Chlamydiaceae* family consists of the single genus *Chlamydia*, which comprises 13 species: *Chlamydia (C.) trachomatis*, *C. pneumoniae*, *C. psittaci*, *C. abortus*, *C. felis*, *C. pecorum*, *C. suis, C. gallinacea, C. caviae, C. avium, C. serpentis, C. poikilothermis* and *C. muridarum* [1]. *Chlamydiaceae* can establish prolonged persistent infections, which are typically asymptomatic, but can also cause disease affecting the eyes, the genital tract, the joints or the respiratory tract, and occasionally cause systemic disease [2]. Chlamydial species can also cause clinical syndromes of variable severity, suggesting that strain and/or host factors may play a major role in disease outcome [3]. In wild and domestic ruminants, *C. abortus*, *C. psittaci*, *C. pecorum* and *C. pneumoniae* are the species most commonly detected in association with disease [4].

*Chlamydia pecorum* is one of the most common species detected in domestic ruminants, associated with reduced weight gain, milk production and conception rates, and therefore having an economic impact on livestock production [5,6]. Clinical signs related to *C. pecorum* infection in ruminants include conjunctivitis, infectious arthritis and spontaneous abortion, as well as encephalomyelitis in cattle [7,8]. *Chlamydia pecorum* has also been detected in wildlife, such as deer [9,10], ungulates from the Caprinae and Bovinae subfamilies [11,12], koala (*Phascolarctos cinereus*) [13] and bird species [14]. Although *Chlamydia pecorum* is a major pathogen in koalas [15], the pathogenic effect of *C. pecorum* in wild hosts is mostly unknown.

*Chlamydia* spp. have occasionally been reported as the main cause of infectious keratoconjunctivitis (IKC) outbreak in bighorn sheep (*Ovis canadensis*) [16], and *C. pecorum* has been involved in an outbreak of ocular disease in reindeer [10]. The clinical signs observed in both outbreaks were ocular discharge, blepharospasm and keratoconjunctivitis. In wild Caprinae, the main etiological cause of IKC is *Mycoplasma conjunctivae* [17], yet it is not clear whether *Chlamydia* spp. has pathogenic synergism in co-infection with *M. conjunctivae* or can cause sporadic ocular disease in wild ruminants. Ocular co-occurrence of *Chlamydia* spp. and *M. conjunctivae* has been reported in chamois, sheep and goats [12,18], and has been associated with a severe IKC outbreak in Pyrenean chamois [19].

The aim of this study is to assess the frequency and diversity of *Chlamydiaceae* in the eyes of wild and domestic ungulates from shared mountain habitats in northern Spain, and to explore the association of *Chlamydia* spp. occurrence with ocular clinical signs, either alone or in co-infection with *M. conjunctivae*.

## 2. Results

*Chlamydiaceae* was detected in seven swab samples from five of the 893 animals tested (0.6%, CI95%: 0.2–1.3). *Chlamydiaceae* DNA was detected bilaterally in the eyes of two chamois. By species, the overall sample prevalence was 0.6% (CI95%: 0.2–1.5) in chamois and 1.4% (CI95%: <0.01–8.1) in sheep. No statistical association was observed between the detection of *Chlamydia* spp. and the animal species. *Chlamydiaceae* were only detected in the eyes of ruminants from the Eastern Pyrenees, but not from the Cantabrian Mountains (Table 1). The sample prevalence of chamois in areas where *Chlamydiaceae* was detected were 0.1% (CI95%: <0.01–0.9) in Alt Pallars National Game Reserve (PAP), 0.1% (CI95%: <0.01–0.9) in Cadí National Game Reserve (PCD) and 0.3% (CI95%: <0.01–1.1) in Freser-Setcases National Game Reserve (PFS). In sheep, *Chlamydiaceae* was only detected in conjunctival swabs from one flock from PAP, with a sample prevalence of 1.4% (CI95%: <0.01–8.1). For both chamois and sheep, the frequency of *Chlamydiaceae* detection was not statistically different between study areas.

*C. pecorum* was identified by the ArrayMate microarray assay in six of the qPCR-positive samples, including bilateral ocular detections in two chamois and unilateral detections in one chamois and one sheep. The identification of chlamydial species was unsuccessful in the remaining qPCR-positive sample from a chamois. No other *Chlamydiaceae* were detected in this study.

Among the sampled ruminants, 63 animals had ocular clinical signs that ranged from mild ocular discharge to corneal perforation [20]. However, none of the *Chlamydiaceae*-positive ruminants were also positive for *M. conjunctivae* or had signs of ocular disease. The detection of *M. conjunctivae* was correlated with the presence of ocular signs, which has been previously published [20].

## 3. Discussion

*Chlamydiaceae* has been associated with ocular disease in wild and domestic animals, yet there is scarce knowledge about the frequency and diversity of chlamydial infections in the eyes of ruminants, and its association with ocular clinical signs. In this cross-sectional study, we found a low frequency of *C. pecorum* in asymptomatic domestic sheep and Pyrenean chamois among the ruminant communities from northern Spain. However, *Chlamydiaceae* may be present in anatomical sites other than the eyes, including the urogenital tract, rectum, joints, brain, lungs or spleen [7,8,21], which may have led to an underestimation of *Chlamydiaceae* infections.

Our results are similar to those reported for the ocular detection of *Chlamydiaceae* in Alpine ibex 1.2% (*Capra ibex*) and Alpine chamois 2.5% (*Rupicapra rupicapra*) from the Swiss European Alps [12]. However, these frequencies are much lower than those reported in both symptomatic and asymptomatic lambs from Australia, ranging from 4% to 73.3% [21,22]. Methodological differences with previous studies prevent a direct comparison of prevalences, since in Australian lambs both serological and different molecular analyses were performed. These methodological differences, such as using flocked vs. non flocked swabs or using different molecular methods, can affect analytical sensitivity for *Chlamydiaceae* detection [23], further complicating prevalence comparison. Apart from the methodological aspects, possible differences in sample prevalence may respond to different epidemiological situations. However, no local differences were detected among our study areas.

*Chlamydia pecorum*, identified in both asymptomatic sheep and chamois in this study, has been previously detected in domestic ruminants presenting syndromes that include ocular disease and polyarthritis [22]. Similarly to our results, previous studies detected *C. pecorum* in the eyes of wildlife inhabiting alpine ecosystems without association with clinical signs, including Alpine ibex [11] and Alpine chamois [12]. Although none of the positive animals of this study had ocular clinical signs, molecular studies of *C. pecorum* suggest that certain strains may be more pathogenic than others [24]. According to experimental infections, animal susceptibility is dose-dependent and may vary in relation to the host’s physiological status and infection route [25]. However, growing evidence indicates that, at least in European Caprinae, *Chlamydia* spp. does not seem to play a major role in outbreaks of ocular-exclusive syndromes [19,20], but has been associated with IKC outbreaks in *Cervidae* worldwide [10,16].

The potential of *Chlamydia* spp. to cause ocular disease may also be affected by co-infections with other pathogens. Microbiological cultures from diseased eyes often yield different bacterial isolates, including *M. conjunctivae*, *Moraxella* spp., *Pseudomonas* spp. or *Staphylococcus* spp. [26]. Although no mixed infections of *Chlamydiaceae* and *M. conjunctivae* were detected in this study, other studies reported *C. pecorum*, *C. abortus* and *C. psittaci* co-infection with *Mycoplasma* species in IKC cases from domestic sheep [18,27], and Alpine chamois [12], suggesting the possibility of interspecies synergism between these pathogens. Arnal et al. also reported a relatively high *Chlamydia* spp. and *M. conjunctivae* ocular co-occurrence in a severe IKC outbreak of Pyrenean chamois [19]. However, *M. conjunctivae* alone is a sufficient cause for IKC in Caprinae [20], and the pathogenic role of *Chlamydia* spp. in these cases of co-occurrence remains unclear. Experimental or longitudinal studies may provide more meaningful insights into the synergism between *Chlamydia* spp. and *Mycoplasma* spp. for the onset and progression of IKC.

The detection of *C. pecorum* in different domestic and wild hosts from alpine ecosystems suggests that transmission between different animal species may occur. However, previous studies reported a great variability between *C. pecorum* strains in co-grazing wildlife and livestock [28], suggesting that these strains may not commonly be transmitted between host species. Further subtyping of the *C. pecorum* strains circulating in chamois and sheep would help to better understand the possible epidemiological links between sympatric hosts from the Pyrenees.

In conclusion, we detected low-frequency *C. pecorum* in the eyes of domestic sheep and Pyrenean chamois from the Eastern Pyrenees, with no relation to ocular disease or evidence of co-infection with *M. conjunctivae*. *Chlamydiaceae* probably did not participate significantly in the pathogenesis of sporadic ocular disease or in the epidemiology of infectious keratoconjunctivitis in the studied mountain habitats from northern Spain. However, further studies are needed to better understand the ecology and pathogenic potential of chlamydial species in wild ruminant hosts and their interface with livestock.

## 4. Materials and Methods

This study was performed from 2009 to 2015 in two areas from the Cantabrian Mountains (NW Spain) and six different areas from the Eastern Pyrenees (NE Spain). The study areas in the Cantabrian Mountains were the National Game Reserves of Mampodre (CMM, 43°01′31” N, 05°11′18” O) and Riaño (CMR, 43°02′56” N, 05°10′40” O). In the Eastern Pyrenees, the study areas were PAP (42°35′09” N, 01°17′06” E), Boumort National Game Reserve (PBM, 42°14′06” N, 01°08′04” E), PCD (42°16′49” N, 01°40′08” E), Cerdanya-Alt Urgell National Game Reserve (PCU, 42°26′45” N, 01°40′54” E), PFS (42°23′37” N, 02°12′42” E) and Vall d’Aran (PVA, 42°48′11” N, 00°47′15” E).

In total, 1786 conjunctival swabs were collected from 893 wild and domestic ruminants by introducing sterile dry cotton swabs to each eye separately. The wildlife sampled were hunted animals sampled during the regular hunting season, and included Pyrenean chamois *(Rupicapra pyrenaica)*, sheep *(Ovis aries)*, roe deer *(Capreolus capreolus)*, red deer *(Cervus elaphus)*, mouflon *(Ovis aries musimon)*, fallow deer *(Dama dama)* and Iberian ibex *(Capra pyrenaica)*. Table 1 summarizes the number of wild ruminants sampled by study area, which is mostly representative of their presence and abundance in each study area. Domestic sheep flocks that seasonally graze the alpine meadows of the study areas were sampled in PVA (one flock, size = 4200), PAP (two flocks, size = 800 and 600) and PFS (one flock, size = 300) (Table 1). Clinical signs and location for each animal were recorded. Swabs were transported at cool temperature in a portable refrigerated box and stored frozen at −20 °C until analyses.

Conjunctival swabs were thawed, cut and mixed during one minute with 0.5 mL of lysis buffer (100 mM Tris-base, pH 8.5, 0.05% Tween 20) in sterile tubes. The lysates were obtained by incubating the tubes at 60 °C for 60 min after adding 0.024 mL of proteinase K. Finally, proteinase K was inactivated at 97 °C for 15 min [29]. DNA was extracted directly from 200 µL of the swab sample lysates using MagAttract 96 *cador* Pathogen Kit (Qiagen, Venlo, Netherlands), following the manufacturer’s instructions.

*Chlamydiaceae* DNA was searched with a SYBR green-based qPCR assay using the primers Chuni-1F (5′-GGG CTA GAC ACG TGA AAC CTA-3′) and Chuni-2R (5′-CCA TGC TTC AAC CTG GTC ATA A-3′) and following previously reported cycling conditions. Briefly, each reaction consisted of 2.5 μL of DNA sample, 12.5 μL of SYBRGreen PCR Master Mix 2x (Applied Biosystems, Warrington, UK), 400 nM of each forward and reverse primer and nuclease-free water to a total volume of 25 μL. *C. psittaci* DNA was used as positive control [30].

The samples positive for the SYBR green-based PCR were sent to the University of Zurich, Switzerland, for confirmation by a family-specific *Chlamydiaceae* real-time PCR *Chlamydiaceae* targeting the 23S rRNA gene using the primers Ch23S-F (5′-CTG AAA CCA GTA GCT TAT AAG CGG T-3′), Ch23S-R (5′- ACC TCG CCG TTT AAC TTA ACT CC-3′) and probe Ch23S-p (FAM-CTCATCATGCAAAAGGCACGCCG-TAMRA) [31]. Chlamydial species identification was performed using a species-specific 23S rRNA gene ArrayMate microarray assay (Abbott, Chicago, IL, USA; Alere Technologies), as established previously [32].

The molecular detection of *M. conjunctivae* in the eyes of the animals included in this study was previously published [33,34], and those results were integrated in this study (data not shown) in order to assess the occurrence of ocular co-infections and possible effects on the onset of clinical signs.

To assess the differences over *Chlamydiaceae* detection according to the area and the animal species of study, Pearson’s Chi-squared test (χ2) was implemented using the R statistical software [35]. Statistical significance was set as *p* < 0.05.

## Figures and Tables

**Table 1 pathogens-10-00383-t001:** Wild and Domestic Ruminants Sampled from 2009 to 2015 in the Cantabrian Mountains (Study Areas CMM and CMR) and the Eastern Pyrenees (study areas of PAP, PBM, PCD, PCU, PFS and PVA). The Sample Prevalence and the 95% Confidence Interval (CI95%) are Only Calculated for Bigger Sample Sizes (>10 Animals).

	C. Mountains		Eastern Pyrenees						*Chlamydiaceae* Prevalence % (Positives/Total)	CI 95%
	CMM	CMR	PAP	PBM	PCD	PCU	PFS	PVA		
Chamois	0/20	0/34	1/77		1/192	0/13	2/262	0/90	0.6 (4/688)	0.2–1.5
Fallow deer			0/5						0.0 (0/5)	NA
Iberian Ibex		0/1							0.0 (0/1)	NA
Mouflon			0/1				0/37		0.0 (0/38)	0.0–10.9
Red deer		0/2	0/4	0/18	0/10	0/1		0/5	0.0 (0/40)	0.0–10.4
Roe deer		0/4	0/9		0/5	0/2	0/19	0/10	0.0 (0/49)	0.0–8.7
Sheep			1/39				0/13	0/20	1.4 (1/72)	<0.01–8.1
All species	20	41	135	18	207	16	331	125	0.6 (5/893)	0.2–1.3

## Data Availability

Data from a previous publication (https://doi.org/10.1371/journal.pone.0186069, accessed on 23 March 2021) about *Mycoplasma* detection in the study samples (not shown) was used to assess co-occurrence in this article. The data generated during and/or analysed during the current study are available from the corresponding author on reasonable request.

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
