# Peer review of "Molecular Detection and Identification of Chlamydiaceae in the Eyes of Wild and Domestic Ruminant Hosts from Northern Spain"

_pathogens, 2021, doi:10.3390/pathogens10030383_

Round 1
Reviewer 1 Report
The study presented looks to identify Chlamydia spp. in both diseased and healthy eyes of domestic sheep and wild ruminants. The samples were collected as eye swabs from many animals hat share mountain habitats from northern Spain. The presence of different Chlamydia spp. was tested for by PCR.
Major concern:
M. conjunctivae was discussed in the introduction and then again in the method section but there was no mention of the data obtained. Specifically, in the last paragraph page 4 and first paragraph page 5: Authors say that molecular detection of M. conjunctivae was done and then integrated into this study. Then the authors proceed to say that the data is not shown?
This reviewer suggests adding this data in since the manuscript is rather light on data and it is being referenced in the materials and methods (as well as introduced in the introduction and discussion). Alternatively the manuscript can be rewritten to remove M. conjunctivae.
Specific comments:
Page 2, second paragraph: italics missing on Chlamydia, C. pecorum, and M. conjunctivae.
Page 2, last paragraph: Chlamydia pecorum should be C. pecorum and Chlamydiaceae is not in italics
Page 4, qPCR methods. Please provide sequences for Chuni-1F and Chuni-2R primers.
Same paragraph as previous comment: Chlamydia psittaci should be C. psittaci
Reviewer 2 Report
The manuscript describes presence and diversity of Chlamydia spp. in ocular specimens from domestic sheep and wild ruminants in northern Spain.
The observed frequency was low, and no association to clinical disease was observed.
The study seems to be well-planned and conducted. The sample size is fairly large and the methods used have earlier been described and validated. Unfortunately (or maybe fortunately), very few positives were found.
My comments are minor:
The title should better reflect the finding (only C. pecorum)
Discussion 4th line: I would say ”…we found a low frequency of…” (see also Conclusions paragraph)
Differences in the methodology are mentioned in Discussion. Could the authors compares the sensitivity (and specificity) of the methods used here and eg., in Australia?
If use of flocked swabs leads to improved detection, why were they not used here?
The discussion would benefit from condensation, and this should be done.
Out of curiosity, what could cause the symptoms in the 63 animals that had ocular signs?
Round 2
Reviewer 1 Report
This reviewer is satisfied with edits and recommends acceptance of the manuscript